**Unleashing the Potential of Geostationary Satellite Observations in Air**
**Quality Forecasting Through Artificial Intelligence Techniques**
Chengxin Zhang[1], Xinhan Niu[1], Hongyu Wu[2], Zhipeng Ding[2], Ka Lok Chan[3], Jhoon Kim[4],
Thomas Wagner[5], Cheng Liu[1,6,7*]
[1]Department of Precision Machinery and Precision Instrumentation, University of Science and
Technology of China, Hefei, 230026, China
[2]School of Environmental Science and Optoelectronic Technology, University of Science and
Technology of China, Hefei, 230026, China
[3]Rutherford Appleton Laboratory Space, Harwell Oxford, United Kingdom
[4]Department of Atmospheric Sciences, Yonsei University, Seoul, Republic of Korea
[5]Satellite Remote Sensing Group, Max Planck Institute for Chemistry, Mainz, Germany
[6]Key Laboratory of Environmental Optics and Technology, Anhui Institute of Optics and Fine
Mechanics, Chinese Academy of Sciences, Hefei, 230031, China
[7]Key Laboratory of Precision Scientific Instrumentation of Anhui Higher Education Institutes,
University of Science and Technology of China, Hefei, 230026, China
*Correspondence: Cheng Liu (chliu81@ustc.edu.cn)
**Abstract.**
Air quality forecasting plays a critical role in mitigating air pollution. However, current
physics-based air pollution predictions encounter challenges in accuracy and spatiotemporal
resolution due to limitations in the understanding of atmospheric physical mechanisms,
observational constraints, and computational capacity. The world's first geostationary satellite
UV-Vis spectrometer, i.e., the Geostationary Environment Monitoring Spectrometer (GEMS),
offers hourly measurements of atmospheric trace gas pollutants at high spatial resolution over
East Asia. In this study, we successfully incorporate Geostationary satellite observations into
a neural network model (GeoNet) to forecast full-coverage surface nitrogen dioxide ($NO_2$)
concentrations over eastern China at 4-hour intervals for the next 24 hours. GeoNet leverages
spatiotemporal series of satellite $NO_2$ observations to capture the intricate relationships among
air quality, meteorology, and emissions in both temporal and spatial domains. Evaluation
against ground-based measurements demonstrates that GeoNet accurately predicts diurnal
variations and spatial distribution details of next-day $NO_2$ pollution, yielding the coefficient of
determination of 0.68 and root mean square of error of 12.31 μg/m$^3$, significantly surpassing
traditional air quality model forecasts. The model's interpretability reveals that geostationary
satellite observations notably improve $NO_2$ forecast capability more than other input features,
especially over polluted regions. Our findings demonstrate the significant potential of
geostationary satellite observations in artificial intelligence-based air quality forecasting, with
implications for early warning of air pollution events and human health exposure.
**Keywords:** air quality forecast; deep learning; health impact; satellite remote sensing;
nitrogen dioxide;

## 1 Introduction

Since the industrial revolution, numerous countries worldwide have encountered severe air pollution issues such as photochemical ozone smog and haze pollution (Hong et al., 2019), which significantly affect human health, crop yields, and the global environment (Manisalidis et al., 2020; Sathe et al., 2021; Guarin et al., 2024). Recent studies have shown that both long-term and short-term exposure to air pollutants such as nitrogen dioxide ($NO_2$) can significantly affect human health, especially the respiratory system (Meng et al., 2021). Accurate and high spatial resolution predictions of air pollutant concentrations can provide critical information for sensitive persons to mitigate health risks. Meanwhile, air quality health risk (AQHI) forecasts and corresponding public response recommendations need to be communicated to the public promptly through public facilities (Tang et al., 2024; Fino et al., 2021). In recent decades, the advancement of atmospheric monitoring and modeling has enabled significant progress in air quality forecasting based on our understanding of atmospheric physics and chemistry (Peuch et al., 2022). Air pollution forecasting not only facilitates responses to environmental health risks but also improves the accuracy of climate and weather simulations (Makar et al., 2015). However, due to our still limited understanding of atmospheric mechanisms and observational and emission constraints, existing air quality forecasts based on physical or statistical models still face challenges in terms of temporal, spatial, and accuracy aspects (Campbell et al., 2022; Zhong et al., 2021).

Artificial Intelligence (AI) technology has made breakthroughs in the field of Earth science (Zhong et al., 2021; Boukabara et al., 2020), particularly excelling in addressing complex problems that are challenging for traditional physical paradigms to simulate (Irrgang et al., 2021), such as weather and climate forecasting (Andersson et al., 2021). Concerning meteorological data, some large-scale deep learning models have surpassed the predictive capabilities of existing numerical weather models to some extent, examples include Climax

(Nguyen et al., 2023), Pangu-Weather (Bi et al., 2023), and GraphCast (Lam et al., 2023).
Despite significant progress and impressive performance achieved in meteorological variables
forecasting with AI methods, there are still limitations in predicting atmospheric pollutant
compositions. Compared to meteorological parameters, the prediction of air pollutant
concentrations is affected by synoptic meteorology, chemistry, and anthropogenic emission
activities, usually with more complex driven mechanisms and associated uncertainties. Current
AI-based air quality forecasts often involve time series predictions at a limited number of
observation stations, rather than full-coverage predictions over the entire spatial domain (Du
et al., 2021). This is primarily due to the lack of effective air quality observations with high
temporal and spatial resolution simultaneously.
While past polar-orbiting satellite observations such as the Ozone Monitoring Instrument
(OMI) and the TROPospheric Monitoring Instrument (TROPOMI), have provided extensive
coverage of atmospheric pollutant distributions such as nitrogen dioxide ($NO_2$), sulfate dioxide
($SO_2$), ozone ($O_3$), and aerosols, they are limited to once-daily overpasses and usually affected
by clouds (Van Geffen et al., 2022; Chan et al., 2023). This frequency usually exceeds the
chemical lifetimes of many reactive gas pollutants like $NO_2$, making it challenging to offer
effective observational constraints for machine learning short-term air quality forecasting
(Shah et al., 2020). However, these observations at a fixed daily overpass time could hardly
support the prediction of atmospheric trace gas concentrations at other times of the day under
different meteorological conditions. In February 2020, the world's first geostationary satellite
payload for air pollution monitoring, the Geostationary Environment Monitoring Spectrometer
(GEMS), began to provide high-coverage and high-precision air quality observations at an
hourly rate for the East Asian region (Kim et al., 2020). The dynamic processes of air pollutants
including emission, transformation, and transport can be observed by the geostationary satellite
during the daytime. This monitoring capability may advance data-driven air quality forecasting
such as machine learning techniques by offering unprecedented observational constraints with
high spatial and temporal coverage. Recent observing system simulation experiments (OSSE)
indicate that assimilating trace gas observations by geostationary satellites into chemical
models can effectively improve surface ozone simulations (Shu et al., 2023), nitrogen oxides
($NO_x$), and emission estimates (Hsu et al., 2024).

Here, based on the unprecedented temporal and spatial resolution and coverage of the

GEMS satellite (Kim et al., 2020), we incorporated Geostationary satellite remote sensing of
tropospheric $NO_2$ column densities (refer to section 4 for details) into a neural Network model
(GeoNet), to forecast full-coverage surface $NO_2$ concentration over the next day from the
current time $t$ (i.e., t+24h). Compared with previous air quality forecasting based on the
simulation of atmospheric physics and chemistry possibly combined with data assimilation
approaches, GeoNet relies solely on geostationary satellite measurements and ancillary
meteorology data. GeoNet effectively addresses the complex nonlinear relationships between
future short-term air quality and current satellite observations, as well as temporally adjacent
meteorological variables (Zhang et al., 2022). The method employs satellite and meteorological
variables within the spatial vicinity of individual air quality monitoring sites as input features,
with site observations serving as labels for model training. The resulting model achieves
accurate and comprehensive air quality predictions across the entire domain over East China,
which is a significant achievement given that past machine learning technologies have relied
on only a few stations or polar-orbiting satellite observations.
**2 Materials and Methods**
**2.1 Geostationary satellite observations of atmospheric $NO_2$**

GEMS is the first UV-Vis spectrometer at a geostationary satellite orbit, measuring

atmospheric pollutants such as $NO_2$, $SO_2$, $O_3$, and HCHO over East Asia, at a spatial resolution
of 3.5 km × 7.5 km at nadir and a temporal resolution of 1 hour during the daytime (Kim et al.,
2020). Based on the unique spectral absorption of trace gases, the atmospheric $NO_2$ column
can be retrieved in visible wavelengths from the spectra of back-scattered sunlight. The details
of the GEMS $NO_2$ retrieval can be found in the Algorithm Theoretical Basis Document
(available at https://nesc.nier.go.kr/ko/html/satellite/doc/doc.do, last access: June 1, 2023). In
this study, we used the tropospheric $NO_2$ column from the GEMS $NO_2$ version 2.0 product, as
well as the cloud fraction for each satellite ground pixel. Overall, GEMS $NO_2$ measurements
have a good correlation with ground-based remote sensing instruments, with correlation
coefficients (R) between 0.69-0.81, and root mean square of errors (RMSE) between 3.2-
$4.9 \times 10^{15}$ *molecules/cm$^2$* (Kim et al., 2023). Our previous validation results indicated that
GEMS $NO_2$ retrievals generally agreed well with ground-based MAX-DOAS measurements
from 6 sites in China, with correlation coefficients ranging between 0.69-0.92 (Li et al., 2023).
**2.2 Ancillary datasets**

Other input information including meteorological datasets is necessary to better constrain

the prediction of future $NO_2$ pollution. Here, both the ERA5 meteorology reanalysis (Hersbach
et al., 2020) and the CAMS forecast (Peuch et al., 2022) were used to provide meteorological
parameters such as zonal and meridional wind (U-wind and V-wind), temperature (Temp),
relative humidity (RH), and precipitation (Precip). In addition, the fraction of cloud cover
available from the satellite $NO_2$ datasets was also considered. To fill the missing gaps in the
satellite $NO_2$ measurements, we use both the $NO_2$ concentrations from the WRF-Chem model
(Zhang et al., 2022) and the CAMS forecast of atmospheric composition. Note that the
reanalysis datasets were typically updated with a week delay from real-time, while the forecast
datasets can provide future 7-day meteorology from the current time. Therefore, the latency of
input datasets would affect the operational prediction of the GeoNet model. Surface $NO_2$
measurements were used as the ground-truth label in the model training phase, available from
over 1000 national air quality sites by the China National Environmental Monitoring Centre
(CNEMC) (Kong et al., 2021).
The preprocessing steps of model input datasets, including outlier detection, missing value
handling, resampling, and normalization, are described in Supplementary Text S1.
**2.3 The GeoNet model**

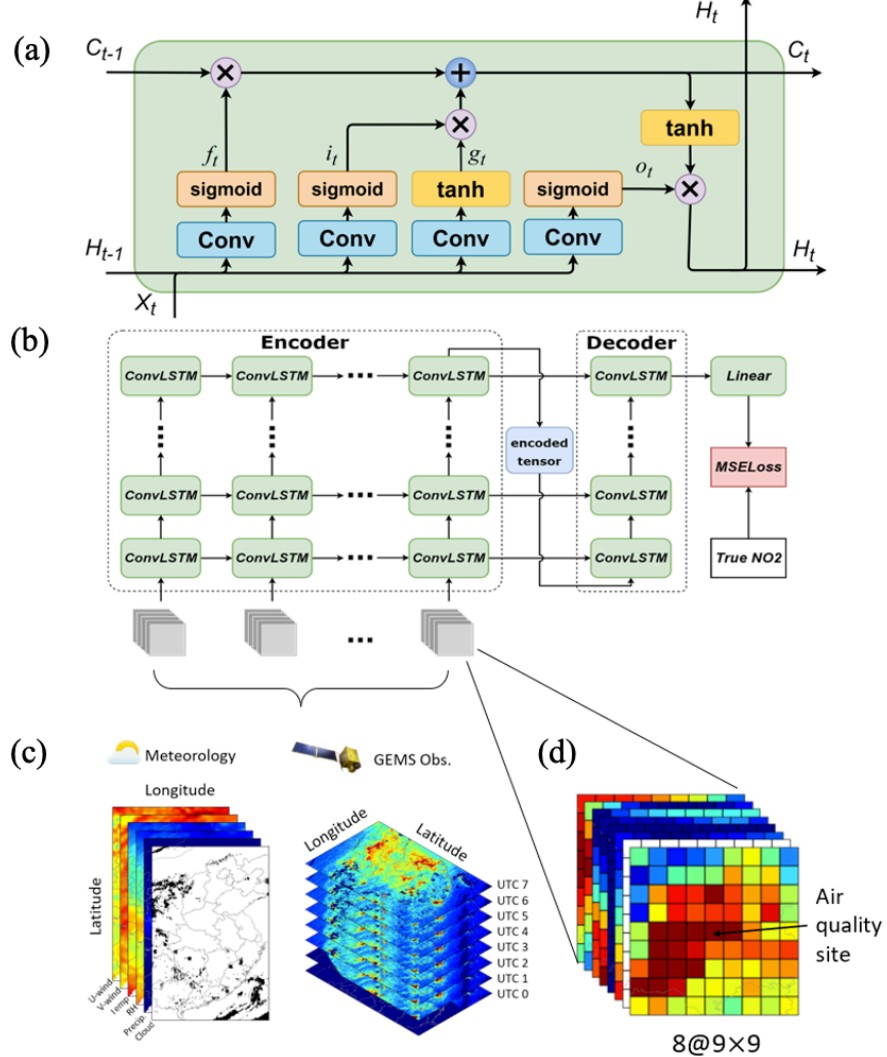

**Figure 1.** The framework of predicting surface $NO_2$ map based on Geostationary satellite measurements and
a ConvLSTM neural network model (GeoNet). (a) the structure of the ConvLSTM block; (b) a diagram of
GeoNet model structure with inputs and output; (c) an illustration of the model input parameters including
meteorological variables and hourly $NO_2$ measurements by the Geostationary satellite; (d) the input data
cube of different features for single training batch, which is centered at an air quality site.
Fig. 1 illustrates the structure and methodology of the artificial intelligence air quality
forecasting model established in this study. Given the distinctive nature of spatiotemporal
sequence data for air quality, predictions must consider not only temporal relationships but also
spatial correlations. The deep learning model employed in this research utilizes convolutional
long short-term memory (ConvLSTM) as its kernel, a variant of the LSTM model designed for
the time series forecasting (Lin et al., 2020). It incorporates a convolutional network structure
to capture spatial features of three-dimensional inputs. Both input-to-state and state-to-state
transitions involve convolutional structures. ConvLSTM determines the future state of a unit
within a grid based on inputs from its local neighbors and past states, allowing it to effectively
model the spatiotemporal dynamics of air quality. The ConvLSTM kernel structure employed
in training is illustrated in Fig. 5a. Here, $X_t$ represents the input at time t, $H_t$ and $H_{t-1}$ denote
the outputs at times t and t-1, and $C_t$ and $C_{t-1}$ represent the states at times t and t-1. The
computational process is as follows:

$$i_t = \sigma(X_t * w_{xi} + H_{t-1} * w_{hi} + b_i) \quad (1)$$

$$f_t = \sigma(X_t * w_{xf} + H_{t-1} * w_{hf} + b_f) \quad (2)$$

$$o_t = \sigma(X_t * w_{xo} + H_{t-1} * w_{ho} + b_o) \quad (3)$$

$$g_t = tanh(X_t * w_{xg} + H_{t-1} * w_{hg} + b_g) \quad (4)$$

$$C_t = f_t \times C_{t-1} + i_t \times g_t \quad (5)$$

$$H_t = o_t \times tanh(C_t) \quad (6)$$

Where the asterisk ($*$) represents the convolution operator, $w$ is the convolution kernel, $b$ is the
offset, $tanh$ is the hyperbolic tangent function, and σ is the activation function of Sigmoid.

The model primarily consists of three components: an encoder, a decoder, and fully

connected layers. Tropospheric $NO_2$ observations from the GEMS satellite for different
consecutive hours within a day, along with corresponding meteorological forecast field data,
serve as input features for model training. The encoder processes the spatiotemporal sequences
of input features for the preceding 8 hours (t-7h, t-6h, …, t), which are then decoded by the
decoder. The final output, representing $NO_2$ concentrations at 4-hour intervals for the next 24
hours (t+4h, t+8h, t+12h,…, t+24h), is produced through fully connected layers. The loss
function of mean squared error (MSE) is calculated by comparing the model output with the
actual values from station observations, and the model undergoes iterative training. In the
training task for a single station sample, the model utilizes continuous and distinct hourly
dynamic images of all variables within the spatiotemporal vicinity of the station as input (see
Fig. 1c-d). This effectively considers the intricate correlations in time and space between air
quality, satellite observations, and meteorological input features. We train the GeoNet model
with input features during the whole year of 2021. The training datasets were randomly selected
from 75% of the whole samples, while the remaining 25% were used as validation sets.
**2.4 The model configuration and optimization**
The model configurations and hyperparameters such as the optimizer, loss function, L1 or
L2 regularization, dropout, training steps, and epochs can make a difference to the model
performance including the prediction accuracy and generalizability. The performance metrics
such as the coefficient of determination ($R^2$), root mean square of error (RMSE), mean absolute
error (MAE), and mean absolute percentage error (MAPE), were used to diagnose the model
(see definition in Supplementary Text S2). Thus, several scenarios of model hyperparameters
have been tested during the model training phase. The model accuracy on validation datasets
and the learning rate curve were used to diagnose the model hyperparameters. The model
parameters mainly include the number of layers and the dimensions of the hidden layers, both
control the model's capacity. If the model capacity is relatively small, underfitting may occur;
overfitting may exist if it is too large. Therefore, selecting an appropriate model capacity is
crucial for improving model performance. During the pre-training process, the model is trained
by combining different numbers of layers and dimensions of the hidden layers. The Mean
Squared Error (MSE) Loss is recorded for each training iteration, and a heatmap is generated
as shown in Fig. S2. From the heatmap, it can be observed that when the number of layers is 2
and the dimension of the hidden layer is 256, the model achieves the minimum MSE Loss. Fig.
S3 shows the sensitivity test results of model loss varying with different batch size settings,
indicating that a batch size of 64 is optimal. Based on the model's MSE loss under different
hyperparameter configurations, the best-fitting model can be selected.
The Adam optimization algorithm controls the learning rate, which can design
independent adaptive learning rates for different parameters. The three initialization parameters
$\epsilon$, $\rho 1$, and $\rho 2$ of the Adam algorithm are set to be 0.0001, 0.9, and 0.99, respectively. For the
epoch, its size is controlled by the early stop method. The early stop method monitors the
change of the model's loss function on the validation set during the training process and stops
the model training immediately when the validation loss of the model starts to become larger.
Due to the fluctuation of the loss function, a threshold $p$ is set for the early stopping method in
practice, and when the validation loss of the model becomes large for $p$ consecutive epochs,
the model is rolled back to the lowest validation loss and the training is stopped, and the
threshold $p$ is set to 10 in this paper. Fig. S4 shows a typical learning curve of the MSE loss in
training and validation data sets for different learning steps in training an optimal model. Such
diagnostics can be used to avoid the model overfitting.
**2.5 The importance of the model input feature**
Permutation feature importance is a technique used to assess the significance of each input
feature in a machine-learning model (Altmann et al., 2010). The core idea is to evaluate the
impact of each feature on model performance by randomly shuffling its values and observing
the resulting change in the model's accuracy. In this study, for each input feature of the GeoNet,
we iteratively shuffle its value independently while keeping other features unchanged, and then
observe the model prediction on the modified input. The difference in the model prediction
performance between using the original and shuffling input quantifies the feature's importance.
Here, we measure the relative importance of each input feature using the metric of 1-$R^2$, due
to its good standardized and indicative ability (Zhang et al., 2022). Generally, a larger
performance drop indicates greater importance, as the model heavily relies on that feature for
predictions. Conversely, smaller drops or increases suggest the feature may be less crucial or
redundant. By permuting the input feature array based on the different spatial and temporal
domains, we can gain a deeper understanding of how feature importance varies spatially and
temporally. For example, the relative importance of one meteorology variable may vary with
different diurnal, weekly, and monthly cycles, revealing the variability of its impact on the
predicted $NO_2$ levels.
**3 Results and Discussion**
**3.1 Model performance**
Based on the GeoNet model and necessary input data (refer to section 2), we have
achieved preliminary predictions of near-surface $NO_2$ concentration with full spatial coverage
and a spatial resolution of 0.1 degrees over eastern China, at four-hour intervals over the next
24 hours. In this study, we first tested the impact of using reanalysis and forecast meteorology
datasets and filling in missing values in satellite observation data on the model predictions. The
reanalysis datasets usually have higher precision than the forecast. Previous studies revealed
that the accuracy of the information on meteorology and chemical composition significantly
affects the performance of machine learning models in estimating air pollutant concentrations
(Zuo et al., 2023; Wang et al., 2024). Due to the shielding effect of clouds, a considerable
proportion of missing values may even exist in satellite $NO_2$ observations. Recent air quality
big-data research usually requires the gap-filling of missing satellite data before inputting it
into the machine learning model, either by spatial interpolation or regression techniques (Kim
et al., 2021). We tested three methods for handling missing data, such as setting them to a fill
value of zero, or replacing them by real-time CAMS simulated $NO_2$, or WRF-Chem simulated
$NO_2$ results (not real-time, but with higher precision).

The comparison results to the validation datasets indicate that the scenario using CAMS

meteorology datasets and replacing missing satellite $NO_2$ data with fill-values (Fig. 2c),

corresponds to a modest $NO_2$ prediction performance with $R^2$=0.68 and RMSE=12.26 μg/m$^3$.

In contrast, the configuration scenario using ERA-5 reanalysis meteorology and imputing with

WRF-Chem simulations (Fig. 2a), corresponds to the best prediction performance of $R^2$=0.69

and RMSE=11.88 μg/m$^3$. This may indicate that the importance of satellite missing data

imputation may be diminished by cloud mask inputs, especially since the model can extract

informative features from spatial and temporal neighboring inputs. To compromise between

the performance of real-time and accuracy, we selected the configuration scenario of using

CAMS meteorology and imputing with CAMS $NO_2$ (Fig. 2d) for subsequent discussion and

operational forecasting, with an $R^2$=0.68 and RMSE=12.31 μg/m$^3$. In summary, the use of

higher-precision meteorology and filling missing $NO_2$ data enhances the model's prediction

accuracy on the validation dataset, but to a rather limited extent. This suggests that, unlike

previous machine learning techniques, GeoNet can effectively adapt to three-dimensional

inputs of varying accuracy and type, fully explore the spatiotemporal correlation of data

features, and demonstrate strong model generalization capabilities.

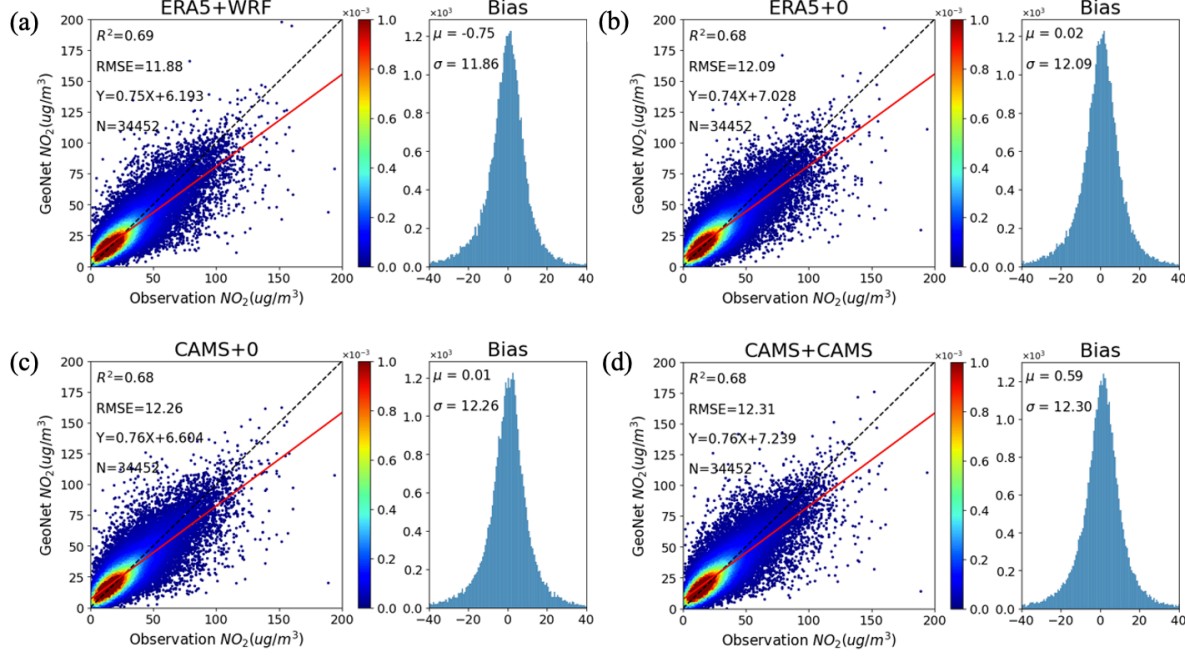


**Figure 2.** The GeoNet prediction performance of the surface $NO_2$ concentration compared to the validation
samples, based on different input datasets of meteorology and atmospheric composition: (**a**) use ERA5
meteorology and fill satellite measurement gaps with WRF-Chem simulated $NO_2$; (**b**) use ERA5
meteorology and $NO_2$ fill-value of zero for over gaps; (**c**) use CAMS meteorology and $NO_2$ fill-value of zero
for gaps; (**d**) use CAMS meteorology and CAMS $NO_2$. The left plot shows the scatter comparisons between
GeoNet predictions and site observations, while the right plot shows the bias distribution between the two.
Figs. S5-S8 provide an overview of the major metrics (e.g., $R^2$, RMSE, MAE, and MPE)
of GeoNet prediction performance varying with prediction hours from t+4h to t+24h in
different months. The results indicate that the model exhibits a higher correlation in $NO_2$
forecast during the spring and winter seasons compared to the summer, while the RMSE errors
show the opposite trend. This could be attributed to much higher $NO_2$ pollution levels in winter
months. Additionally, GeoNet's $NO_2$ prediction errors gradually increase during the next 24
hours, particularly after t+20h. This is primarily due to the short lifetime of atmospheric $NO_2$,
leading to a diminishing constraint from historical observational data on future $NO_2$ predictions.
Similar phenomena are also observed in machine learning or model-assisted weather forecasts
(Andersson et al., 2021).
To assess the GeoNet model's performance for short-term pollution events, we compared
it with near-surface $NO_2$ from CAMS forecasts, and in situ observations from CNEMC ground
stations. Fig. S9 illustrates the daily time series of t+4h $NO_2$ from GeoNet, CAMS, and
CNEMC for three typical sites in Beijing, Shanghai, and Guangzhou in 2021. As shown from
the plot, $NO_2$ predictions by both GeoNet and CAMS generally agreed with the variation trends
of CNEMC measurement. However, CAMS forecasts systematically overestimate the surface
$NO_2$ concentration by 100%, possibly resulting from the biases in the $NO_x$ emission inventory
(Douros et al., 2023). Compared to CAMS, the GeoNet prediction closely aligns with the
ground-truth observations at CNEMC sites over eastern China, with an overall $R^2 > 0.5$ and
mean bias $< 5$ μg/m$^3$ for polluted regions (see Fig. S10 and S11, respectively).
**3.2 Main factors in $NO_2$ forecast and their implications**
Previous physics-based numeric models of air quality prediction, e.g., the CAMS global
forecast model and the regional WRF-CMAQ model (Liu et al., 2023; Kumar et al., 2021;
Kuhn et al., 2024), can simulate the atmospheric physical and chemical processes (such as
advection, diffusion, deposition, and chemical reactions) by solving the atmospheric equations.
Recent data assimilation techniques further take real-time monitoring data from satellite and
ground-based platforms as model constraints to better predict air quality variables (Inness et
al., 2022). Compared with physics-based models, "black-box" models such as the deep learning
technique usually lack interpretability and explainability (Zhang and Zhu, 2018). This hinders
the understanding and implications for predicting air quality variables such as $NO_2$. Here, we
measure the relative importance of each input feature on the $NO_2$ forecast accuracy, by
iteratively permuting the input array and observing its influences on the model prediction.

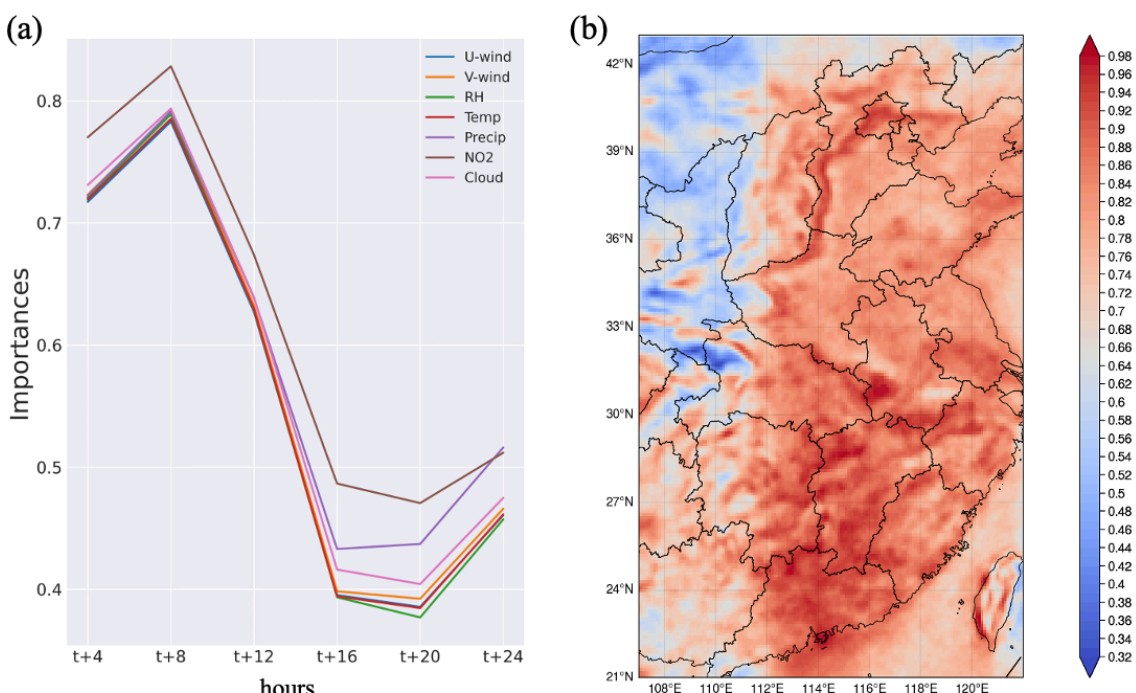


**Figure 3.** (a) The overall relative importance of different input features such as wind, surface pressure,
satellite $NO_2$, and cloud mask, in GeoNet $NO_2$ forecast, varying with different hour steps from t+4h to t+24h.
(b) The spatial distribution of the relative importance of satellite $NO_2$ measurements in the GeoNet $NO_2$
forecast in 2021.
Fig. 3a presents the relative importance $(1-R^2)$ of different input features varying with
prediction hour steps from t+4h to t+24h. The geostationary satellite $NO_2$ measurements play
the highest role in predicting surface $NO_2$ levels of the next day, although it degrades after t+8h.
Other meteorological input features also show a major impact on $NO_2$ prediction performance.
The significance of the different input variables remained generally consistent across seasons,
with minor variations (as shown in Fig. S12). By permutating the input array for each ground
pixel, Fig. 3b derived the spatial distribution of the relative importance of geostationary satellite
$NO_2$ in the predicting performance. Overall, satellite $NO_2$ has a higher impact in densely
populated areas experiencing severe air pollution, such as the Pearl River Delta, Yangtze River
Delta, and Jianghuai Plain, than in western China. Such results highlight the underappreciated
role of satellite $NO_2$ measurements with high spatial and temporal coverage in air pollution
forecasts.
**3.3 $NO_2$ pollution episodes and health exposure forecast**

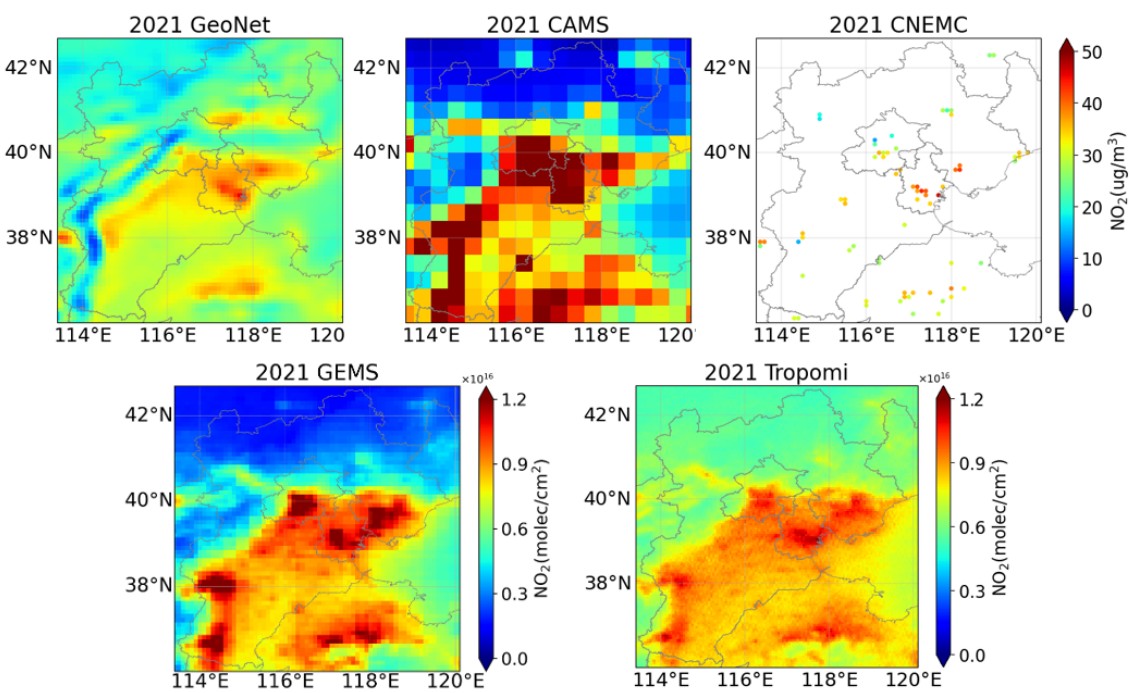

**Figure 4.** The comparisons of annual surface $NO_2$ concentrations from GeoNet, CAMS, and CNEMC,
respectively, (in the top panel), as well as the tropospheric $NO_2$ column observations from GEMS and
TROPOMI over East China in 2021 (in the bottom panel).
Beyond its prediction accuracy, GeoNet exhibits a pronounced advantage in spatial
coverage and resolution, allowing for capturing finer-scale details in the pollutant distribution.
Illustrated in Fig. 4, GeoNet demonstrates remarkable performance in predicting spatial
nuances of NO$_2$ pollution, particularly when contrasted with ground-based and satellite
observations. During a typical winter NO$_2$ pollution event (as shown in Fig. 5), GeoNet
accurately simulates a significant decrease in concentrations at 11:00 and 15:00, probably led
by intense photochemical activity in the daytime, coincident with ground-based observations.
It also outperforms CAMS in predicting NO$_2$ variations throughout the day. The GeoNet model
also retains the distributional differences in NO$_2$ concentrations between urban and rural areas,
consistent with emission source characteristics and satellite observations. The suboptimal
performance of CAMS predictions can be attributed to insufficient observational constraints
and the use of outdated emission inventories (Douros et al., 2023). In the European region, the
assimilation of TROPOMI observations into CAMS forecasts significantly improves the
simulation accuracy of near-surface NO$_2$ concentrations and tropospheric column densities
(Inness et al., 2019). Neural network methods, similar to GeoNet, could be used to correct and
downscale forecast results by existing models (Baghanam et al., 2024). This approach holds
promise for achieving operational air quality forecasts that balance efficiency and accuracy.

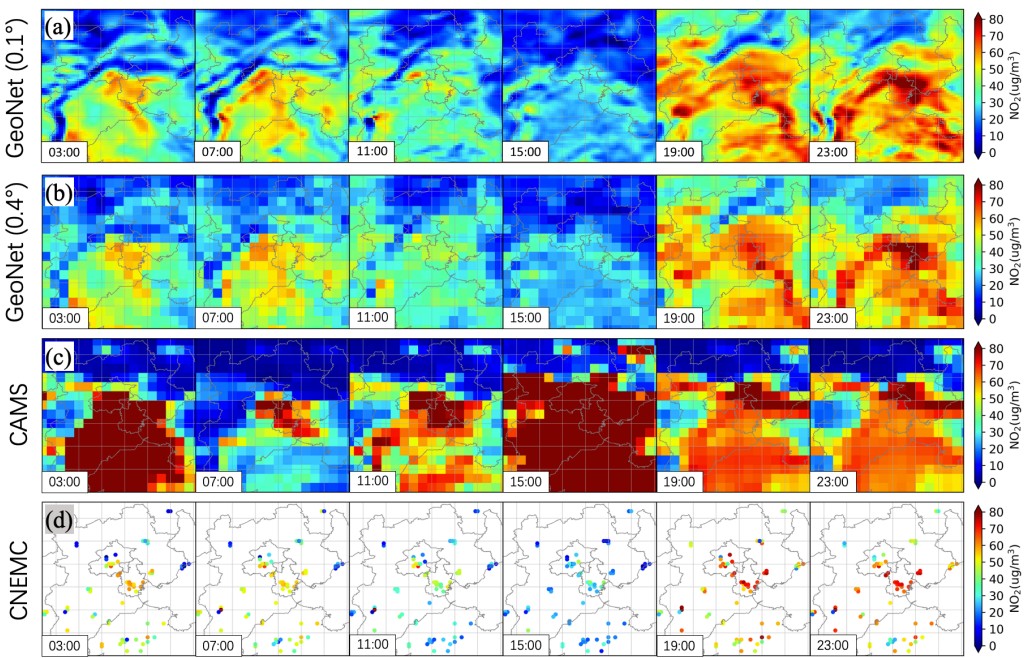


**Figure 5.** The spatial distribution comparisons of surface NO$_2$ concentration between (a) GeoNet prediction at the original resolution of 0.1°, (b) GeoNet prediction resampled to the CAMS resolution of 0.4°, (c) CAMS prediction, and (d) ground-based CNEMC site measurements. Note that the results are presented for different continuing local hours (labeled text in the subplot) on 23 November 2021.

In this study, we used a simplified linearized risk model for the short-term NO$_2$ exposure (Meng et al., 2021; Zhang et al., 2022) to calculate the distribution of all-cause mortality risks based on GeoNet NO$_2$ predictions (see Fig. 6). Short-term NO$_2$ exposure leads to remarkable regional differences in all-cause mortality, which are mainly concentrated in highly polluted and densely populated urban areas. For both urban and suburban locations in Beijing (see Fig. 6c-d), GeoNet-based NO$_2$ pollution exposure predictions are more consistent with actual in situ observations than the CAMS forecasts. Current air quality health indices forecasting based on limited station data has significant gaps, making it difficult to meet the refined needs for different populations in urban, suburban, and rural areas. Integrating GeoNet forecasts based on hourly geostationary satellite observations can support spatially comprehensive and fine-scale air quality health risk prediction. This, in turn, guides managing the risks of air pollution exposure-related diseases in sensitive populations and communities.

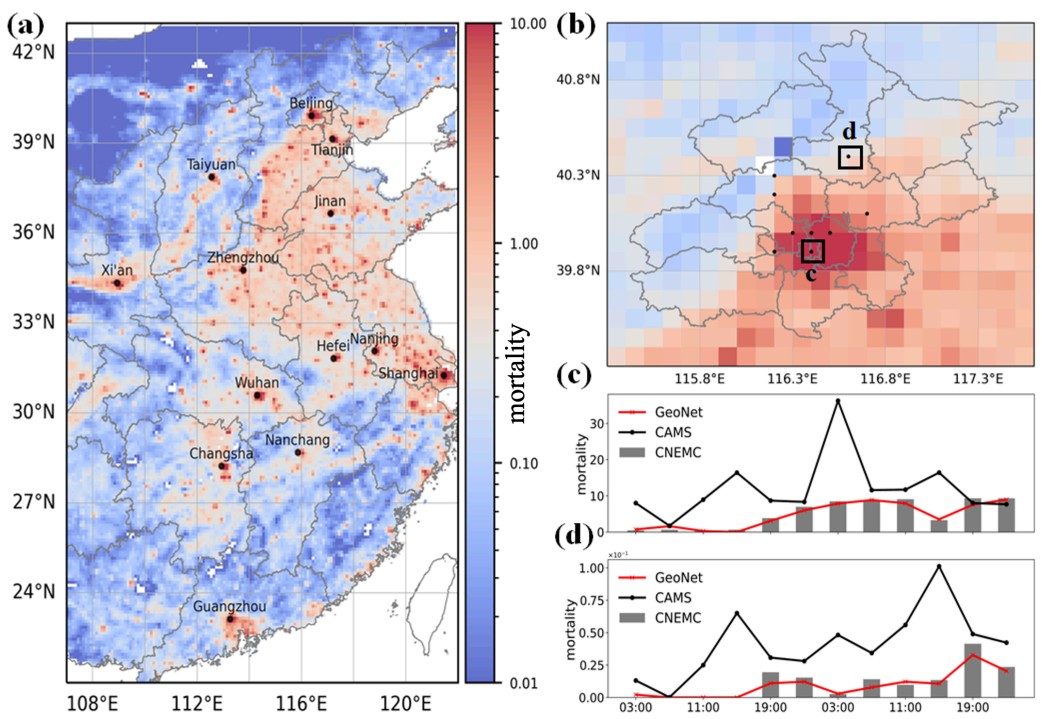

**Figure 6.** Mortality risk of short-term $NO_2$ exposure based on the GeoNet prediction on November 23, 2021. (**a**) mean mortality due to the predicted $NO_2$ exposure in East China; (**b**) a zoom-in map over Beijing and its neighboring area; (**c**) and (**d**) are comparisons of mortality estimation over the Beijing urban and rural regions (the rectangle areas presented in **b**), respectively, based on different $NO_2$ exposure prediction among GeoNet, CAMS, and CNEMC.

**4 Conclusion**

The GeoNet model utilizes the unprecedented hourly air quality observations from geostationary satellites and resolves nonlinear associations in spatiotemporal proximity across multiple data sources. It achieves seamless short-term regional air quality predictions, exhibiting significant performance advantages over existing machine-learning air quality prediction models. To strike a balance between real-time and accuracy requirements, we evaluated the impact of using reanalysis- and forecast-based meteorology datasets, as well as imputing the missing values of satellite $NO_2$. The findings reveal that the GeoNet model demonstrates robust generalization across diverse datasets, with minimal fluctuations in prediction performance. Overall, the model achieves an RMSE of 12.31 μg/m$^3$ and an $R^2$ of 0.68 in predicting $NO_2$ concentrations every 4 hours for the next 24 hours. However, validation accuracy notably diminishes after t+16h within the next 24 hours, with stronger predictive correlations observed in seasons characterized by severe pollution, such as spring and winter, compared to summer. The variation of the model forecasting performance also shows that accurate prediction for longer time windows and heavy pollution events is still a major difficulty. This may be due to the high level of uncertainty in emissions and meteorology. In the future, a combination of higher resolution and more accurate multi-source data constraints, as well as machine learning models coupled with atmospheric physical mechanisms, may be needed to improve the existing forecasts.

Compared to traditional chemical model forecasts and data assimilation predictions, the GeoNet model handles various data sources, including meteorological simulations and air quality observations, and more accurately captures spatial intricacies of air pollution evolution.

The GeoNet framework elucidated in this study forecasts short-term near-surface $NO_2$
concentrations and demonstrates transferable learning potentials for predicting other pollutants.
This work also has important implications for the prediction of near-surface $O_3$ and particulate
matter. For example, the integration of using vertical $O_3$ profiles from the GEMS satellite, in
particular near-surface layer concentrations, and their joint observations of important $O_3$
precursors including $NO_2$ and HCHO, is expected to significantly improve the uncertainty of
existing estimates of near-surface air pollution. This study underscores the pivotal role of next-
generation stationary satellite observations of air pollution constituents in air quality
forecasting, with the potential to advance operational air quality forecasting and mitigate
associated health risks by integrating machine learning technologies.

**Data and code availability.** The GEMS $NO_2$ v2.0 data is available from the National Institute of Environmental Research (NIER) of South Korea (https://nesc.nier.go.kr/en/html/index.do, last access: December 10, 2023). We downloaded the $NO_2$ measurements from the CNEMC real-time air quality platform (https://air.cnemc.cn:18007/, last access: Jun 8, 2023). ERA-5 reanalysis meteorological data is obtained from the European Center for Medium-Range Weather Forecasts (https://climate.copernicus.eu/climate-reanalysis, last access: December 8, 2023). CAMS forecast of meteorological and atmospheric $NO_2$ datasets are retrieved from the CAMS Atmosphere Data Store (https://ads.atmosphere.copernicus.eu/, last access: December 8, 2023). The source codes of the GeoNet model, surface $NO_2$ prediction, and necessary input data can be obtained from Chengxin Zhang (zcx2011@ustc.edu.cn) upon reasonable request.

**Contributions:** C.Z. implemented the GeoNet model and analyzed the data. C.L. supervised the study. C.Z. wrote the manuscript with input from all co-authors.

**Competing interests:** The contact author has declared that none of the authors has any competing interests.

**Acknowledgments.** This study was supported by the National Natural Science Foundation of China (Nos. 42225504, 62305322, and 42375120), the National Key Research and Development Program of China (Nos. 2022YFC3700100 and 2023YFC3706104), the Fundamental Research Funds for the Central Universities (Nos. YD2090002021 and WK2090000038) and the New Cornerstone Science Foundation through the XPLORER PRIZE (2023-1033).

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
