# Peer review of "Unleashing the Potential of Geostationary Satellite Observations in Air"

_EGUsphere, 2024_

## Author Comment (AC1)

**Response to reviewers**

We thank the two reviewers for their thorough and constructive reviews of this manuscript. We have incorporated most of the suggestions and believe the revised paper is substantially improved. Please find below the author's responses point-by-point. The original comments by the reviewer are in *blue italics*, our responses are in black. The revised manuscript with tracked changes is also appended.

**Reviewer #1:**

*Major Comments*
*This study by Zhang et al. entitled "Unleashing the Potential of Geostationary Satellite Observations in Air Quality Forecasting Through Artificial Intelligence Techniques" presents a new machine-learning framework – GeoNet – that synthesizes geostationary observations of columnar NO2 from the Geostationary Environment Monitoring Spectrometer (GEMS) with meteorological parameters to forecast surface-level NO2 in East China. Overall, this study represents a significant advancement in surface-level pollution forecasting given its use of the unprecedented hourly data provided by GEMS. I believe that this manuscript is well-written and consistent; however, I have a few comments below.*

We appreciate the reviewer for the positive comments. We have addressed the following concerns point-by-point below:

1. *First, if possible, it would be useful to validate the GEMS observations using ground-based spectrometers (e.g., PGN) specifically for the study region and time period.*

We thank the reviewer for this suggestion! In our previous work (e.g., (Li et al., 2023)), we validated the GEMS $NO_2$ retrievals with the six ground-based MAX-DOAS distributed over the Jing-Jin-Ji region, Yangtze River Delta, Pearl River Delta, and Sichuan-Chongqing Basin in China. Generally, the correlation coefficient between GEMS and MAX-DOAS $NO_2$ retrieval ranges between 0.69-0.92 for different sites (see Fig. R1), indicating high consistency between both datasets. We have now discussed these results in L125-127, section 2.1:

> *Our previous validation results indicated that GEMS $NO_2$ retrievals generally agreed well with ground-based MAX-DOAS measurements from 6 sites in China, with correlation coefficients ranging between 0.69-0.92 (Li et al., 2023).*

[Figure]

Fig. R1 (from Li et al., 2023). The comparison between GEMS and the MAX-DOAS NO₂ VCDs data from different stations. Panels (a-1), (b-1), (c-1), and (d-1) represent the comparisons with CAMS, GIG, HNU, and CQ stations, respectively. Panel (a-2), (b-2), (c-2), and (d-2) represent time series plots with CAMS, GIG, HNU, and CQ stations, respectively.

2. *Additionally, unless I missed it, I don't believe the time periods for model training and validation were ever stated; if this is the case they should be added to the main text.*

We thank the reviewer for this helpful reminder! We now add the following descriptions in L185-187, section 2.3:

> *We train the GeoNet model with input features during the whole year of 2021. The training datasets were randomly selected from 75% of the whole samples, while the remaining 25% were used as validation sets.*

3. *Second, when investigating feature importance, it would be useful to also identify variability in the feature importance to uncover whether some components are more stable than others in GeoNet and to identify if the significance of geostationary observations is consistent across different days and seasons.*

We now investigated the temporal variations of these features' importance across different seasons. Fig. R2 is similar to Fig. 3a in the main text, but for the feature importance in Spring, Summer, Autumn, and Winter. We added the following discussions in L321-322:

*The significance of the different input variables remained generally consistent across seasons, with minor variations (as shown in Fig. S12).*

[Figure]

Fig. R2 (also moved into the Supplementary Information, Fig. S12). Similar to Fig 3a, but for different seasons, including Spring (a), Summer (b), Autumn (c), and Winter (d).

4. *Lastly, I suggest that the authors update their analysis in Figure 4 to include the GeoNet predictions regridded to the CAMS grid to identify how much of the improvement in predictions is attributable specifically to enhancements in spatial resolution.*

Thanks for this suggestion! We now re-grid the GeoNet predictions to the CAMS grid in Fig. R3 (and also replaced it with Fig. 5). It can be seen that GeoNet prediction results between the original (0.1°) and CAMS resolution (0.4°) are similar in spatial patterns overall.

[Figure]

**Fig. R3** (Fig. 5 in the revised manuscript). The spatial distribution comparisons of surface NO₂ concentration between (a) GeoNet prediction at the original resolution of 0.1°, (b) GeoNet prediction resampled to the CAMS resolution of 0.4°, (c) CAMS prediction, and (d) ground-based CNEMC site measurements. Note that the results are presented for different continuing local hours (labeled text in the subplot) on 23 November 2021.

*I have included line-specific comments below:*
*Minor Comments*
*1. L53-54: While I agree with this statement, it should be mentioned that for air pollution forecasting to facilitate health benefits, infrastructure needs to be created that communicate risks and appropriate responses to risks to the public.*

We added the following lines to mention the necessity of implementing infrastructure to communicate air quality risks and public recommendations, in L50-52 of the revised manuscript:

> Meanwhile, air quality health risk (AQHI) forecasts and corresponding public response recommendations need to be communicated to the public promptly through public facilities (Fino et al., 2021; Tang et al., 2024).

*2. L55: I think you can drop the second limited in this line.*

Removed.

*3. L75: Maybe it would be useful to give an example or two here (i.e., TROPOMI + OMI).*

Added "such as the Ozone Monitoring Instrument (OMI) and the TROPospheric Monitoring Instrument (TROPOMI)".

4. *L78-81: Another limitation of the polar orbiting satellites that is worth mentioning is that typically (at least in the case of TROPOMI) the satellite observes at roughly the same time of day (early afternoon) which makes it difficult to predict concentrations at other times of the day with different meteorological (boundary layer height) and photochemical conditions.*

Thanks for your suggestion. We added the following sentences in L84-86:
   *However, these observations at a fixed daily overpass time could hardly support the prediction of atmospheric trace gas concentrations at other times of the day under different meteorological conditions.*

5. *L92: It would be better to describe GEMS as having "unprecedented temporal and spatial resolution and coverage" as ground-level monitors can observe hourly NO2 but are limited in time and aircraft remote-sensing can observe NO2 at sub hourly resolution but over a limited temporal coverage (usually a few days or weeks). The resolution alone isn't necessarily unique but rather than combined spatial + temporal resolution with extended spatial and temporal coverage.*

Corrected.

6. *L117-120: Were you able to validate these data for the study time period / domain? If possible, it may be useful to compare GEMS to ground-based spectrometers in the study domain to get an idea of performance.*

Please refer to the response to major comment #1.

7. *L207-208: I don't think you need this sentence as it is already mentioned in the methods section.*

Removed.

8. *Figure 3: It would be interesting to present the variance of these different components as well in a). Are these importance values pretty consistent regardless of season and day, or do they vary substantially day to day?*

Please refer to the response to major comment #3.

9. *Figure 4: Have you assessed how much of the reductions in performance are attributable to resolution? If not, I suggest regridding the GeoNet prediction to the resolution of CAMS and comparing this "GeoNET_coarse" product to the observations to characterize how much of the improved performance is attributable to enhanced spatial resolution.*

Please refer to the response to major comment #4.

10. *Figure 5: The colorbar in a is not labeled, and throughout the font is small (especially in the yaxis of c and d), I suggest updating to improve readability.*

Thanks for your helpful suggestion! We added the figure label and increased the font as follows.

[Figure]

**Fig R4 (Fig. 6 in the manuscript).** Mortality risk of short-term NO₂ exposure based on the GeoNet prediction on November 23, 2021. (**a**) mean mortality due to the predicted NO₂ exposure in East China; (**b**) a zoom-in map over Beijing and its neighboring area; (**c**) and (**d**) are comparisons of mortality estimation over the Beijing urban and rural regions (the rectangle areas presented in **b**), respectively, based on different NO₂ exposure prediction among GeoNet, CAMS, and CNEMC.

*11. L338-339: I don't believe the timeframe of this study was mentioned at all in the main text. What months / years was this prediction trained on and for what period was it validated?*

Please refer to the response to major comment #2.

**Reviewer #2:**

*The authors attempted to improve the short-term prediction of surface NO2 at a high spatial and temporal resolution by taking advantage of the GEMS NO2 prodcuts and a neural network model. They successfully forecasted full-coverage surface NO2 for the next 24 hours and identifed the critical role of GEMS NO2. Their results demonstrate the potential application of the GEMS products in air quality prediction.*
*Overall, this is an important study and the results presented here will be useful for future applications of GEMS products as well as the geostationary satellite observations. I am happy to see its publication in due course. However, before that, I still have a few concerns or suggestions for the authors.*

We appreciate the reviewer for the positive comments. We have addressed the following concerns point-by-point below:

1. *I would sugget to move the model configuration and optimization into the main text. This will be very helpful for readers to understand the model.*

Thanks for your suggestion. We now moved the model configuration and optimization details to the L188-219 (Section 2.4) of the main text. Please refer to the details in the revised manuscript.

2. *In the handling of missing data, the authors tried to set them to a fill value of zero. Is it reasonable? It looks reasonable to fill values of diurnal NO2 climatology (e.g., the seasonal mean diurnal NO2). In addition, as shown in Fig.2, it looks the three methods of handling missing data perform similarly in term of R2 and RMSE. So I don't think it is necessary to highlight the "weakest" or "strongest" configuration.*

We greatly appreciate the reviewer's concerns regarding our method of handling missing data.

The reason we chose to use a direct fill value of zero is that during the experiments, we observed that this approach, combined with the model's feature extraction capability, could effectively mitigate the negative impact of missing data on prediction performance. Meanwhile, by incorporating cloud cover (mainly effectors of the missing satellite measurements), this method allows the model to fully leverage its potential, overcoming the issues caused by missing data. We acknowledge the reasonableness of using diurnal NO$_2$ climatology, as the reviewer suggested. However, the data discontinuities between satellite NO$_2$ measurements and gap-filled NO$_2$ climatology could lead to systematical biases in model predictions. Moreover, as indicated by the results, R$^2$ and RMSE metrics did not show significant differences across the various methods of missing data handling. Therefore, we believe our current approach is valid, with minimum effect on data discontinuities.

In general, we now provide a more detailed discussion of the different missing data handling techniques, clarifying the rationale for our choice and highlighting the positive role that cloud-cover data plays in enhancing the model's performance. Please refer to the detailed revisions in L256-263 of the manuscript:

> *The comparison results to the validation datasets indicate that the scenario using CAMS meteorology datasets and replacing missing satellite NO$_2$ data with fill-values (Fig. 2c), corresponds to a modest NO$_2$ prediction performance with R$^2$=0.68 and RMSE=12.26 μg/m$^3$. In contrast, the configuration scenario using ERA-5 reanalysis meteorology and imputing with WRF-Chem simulations (Fig. 2a), corresponds to the best prediction*

*performance of R²=0.69 and RMSE=11.88 μg/m³. This may indicate that the importance of satellite missing data imputation may be diminished by cloud mask inputs, especially since the model can extract informative features from spatial and temporal neighboring inputs.*

3. *I would also suggest to move Fig.S12 in the main text which shows the advantage of GEMS measurements.*

Done. We moved Fig. S12 into the main text (Fig. 4).

4. *The authors also show that the performance of GeoNet model degrades notably after t+16h. Is there any possible solution to overcome this short predicability?*

We appreciate the valuable feedback regarding the performance degradation of our short-term air quality forecasting model beyond t+16h. We acknowledge that this is a common phenomenon in many air quality prediction models, particularly due to the inherent uncertainties in meteorology, emissions, and chemical reactions that significantly affect longer-term forecasts. Although the accuracy of predictions after t+16h declines, the results still provide meaningful insights. In future work, we aim to improve long-term forecasting performance by adopting more sophisticated model architectures and incorporating enhanced observational data or physical constraints.

In response to this comment, we also revised the manuscript in the following aspects:

(1) Explicitly discuss the reasons for the model's performance degradation beyond t+16h in the manuscript.
(2) Outline future research directions, including the exploration of model frameworks with better temporal generalization and the integration of stronger observational and physical constraints to improve long-term predictions.

Please refer to L387-392 in the revised manuscript:

> *The variation of the model forecasting performance also shows that accurate prediction for longer time windows and heavy pollution events is still a major difficulty. This may be due to the high level of uncertainty in emissions and meteorology. In the future, a combination of higher resolution and more accurate multi-source data constraints, as well as machine learning models coupled with atmospheric physical mechanisms, may be needed to improve the existing forecasts.*

5. *As mentioned in my last comment, the authors also highlight the possible applications to other air pollutants. However, the chemistry and lifetime of other air pollutants might be very different from NO2. For example, if the GEMS tropospheric ozone measurement is useful for the prediction of surface ozone? Some more detailed discussions on the possible application to other pollutants would be very insightful.*

Thanks for your comment. We elaborated it in the L398-402 in the revised manuscript:

> *This work also has important implications for the prediction of near-surface O3 and particulate matter. For example, the integration of using vertical O3 profiles from the GEMS satellite, in particular near-surface layer concentrations, and their joint observations of important O3 precursors including NO2 and HCHO, is expected to significantly improve the uncertainty of existing estimates of near-surface air pollution.*

*6. BTW, I am not sure whether it is necessary to include NO2 in the Title of this manuscript since the authors didn't talk too much about major air pollutants (i.e., PM2.5 and ozone)*

Thanks for your comment. Considering the broad implications of this research for forecasting other pollutants using Geostationary satellite measurement (also discussed in response to comment #5), we think it would be better to retain the current title.

**References:**

[revised manuscript text omitted]